# Requirements and Architecture of a Cloud Based Insomnia Therapy and Diagnosis Platform: A Smart Cities Approach

**Daniel Reichenpfader \*** and **Sten Hanke**

Institute of eHealth, FH Joanneum University of Applied Sciences, Eckertstraße 30i, 8020 Graz, Austria; sten.hanke@fh-joanneum.at
**\*** Correspondence: reichenpfader.daniel@gmail.com

**Abstract:** Insomnia is the most common sleep disorder worldwide. Its effects generate economic costs in the millions but could be effectively reduced using digitally provisioned cognitive behavioural therapy. However, traditional acquisition and maintenance of the necessary technical infrastructure requires high financial and personnel expenses. Sleep analysis is still mostly done in artificial settings in clinical environments. Nevertheless, innovative IT infrastructure, such as mHealth and cloud service solutions for home monitoring, are available and allow context-aware service provision following the Smart Cities paradigm. This paper aims to conceptualise a digital, cloud-based platform with context-aware data storage that supports diagnosis and therapy of non-organic insomnia. In a first step, requirements needed for a remote diagnosis, therapy, and monitoring system are identified. Then, the software architecture is drafted based on the above mentioned requirements. Lastly, an implementation concept of the software architecture is proposed through selecting and combining eleven cloud computing services. This paper shows how treatment and diagnosis of a common medical issue could be supported effectively and cost-efficiently by utilising state-of-the-art technology. The paper demonstrates the relevance of context-aware data collection and disease understanding as well as the requirements regarding health service provision in a Smart Cities context. In contrast to existing systems, we provide a cloud-based and requirement-driven reference architecture. The applied methodology can be used for the development, design, and evaluation of other remote and context-aware diagnosis and therapy systems. Considerations of additional aspects regarding cost, methods for data analytics as well as general data security and safety are discussed.

**Keywords:** mHealth; cloud computing; ESM/EMA; mIoT; telemedicine; Smart Cities

## 1. Introduction

Cloud computing, i.e., the storage and processing of data on the internet, is currently experiencing an upsurge: infrastructure-, platform-, and software-as-a-service products (IaaS, Paas, Saas) make the acquisition and maintenance of on-premises Information Technology (IT) infrastructure obsolete and promise cost-effective, foolproof, and compliant data processing worldwide. As in other sectors, cloud computing paves the way for new opportunities in healthcare as well: electronic patient records, telemedicine, and mHealth are among the first applications already implemented based on cloud computing [1–3]. By combining data from different data sources (including environmental and other data) the context-aware health paradigm of Smart Cities could lead to an improved diagnosis of sleep disorders [4]. However, its clinical application requires improved quality criteria of the measurement methods [5–8]. Context-aware health services include standard data from structured domains (diagnoses, procedures, laboratory results, vital signs, medication), patient-reported outcomes (PROs), also called patient-generated health data (PGHD), as well as health related environmental factors and social determinants. Solanas et al. even introduce the concept of s-Health which describe the context-aware complementation of

mobile health within smart cities [4]. The aim of s-Health is to provide citizens and patients with healthcare applications and services that automatically adapt to discovered context by changing these systems' behaviour. Furthermore, it has been shown that insomnia, for example, is related to environmental factors like noise, air, and light pollution which could play a part when diagnosing or treating the disease [9–11].

The improvement of cloud and IT technology is providing more and more possibilities, as well as innovative therapy and diagnosis platforms, taking all the above-mentioned aspects into account. The current paper aims to provide a process of requirement collection based on a concrete application, drafting of a reference architecture, as well as an implementation concept, which can serve as a blueprint for cloud based therapy and diagnosis platform applications. The example chosen in this paper is insomnia as an ICT based application would benefit a lot from the aspects mentioned above (cloud-based data analytics, integrated behaviour data and a diverse dataset, mHealth setup, web service integration for professionals, etc.

Insomnia and its associated complications in falling and staying asleep are among the most common sleep disorders worldwide [12]. Impairments caused by insomnia during the day result in high economic costs, which could be reduced through effective therapy. However, unclear or deviating criteria of various diagnostic manuals impede the distinction of insomnia from other sleep disorders. Cognitive behavioural therapy tailored to insomnia (CBT-I) is considered as the standard procedure for treating insomnia [13]. If diagnostic criteria are uncertain, it is necessary to exclude physiological disorders, such as obstructive sleep apnoea syndrome. In that case, polysomnography (PSG) is needed for clarification: PSG has been the norm the assessment of sleep disorders for over 50 years [14]. Like individual psychological interviews, PSG is expensive, requiring continuous monitoring and manual interpretation of results by trained and certified medical staff. However, the validity of the results of a single night's examination might be low due to the unfamiliar environment of a sleep laboratory. This is also known as the "first-night effect" [15]. The high expenditure of resources and the associated costs could be reduced by digitally supporting the diagnosis and therapy process.

Using non-organic insomnia as an example, the following paper will demonstrate a detailed process, from requirements to an architecture implementation. As a first objective, various requirements for such a platform should be defined comprehensively by means of integrative literature research. Second, a software architecture shall be drafted based on these identified requirements. Last, an implementation concept of the platform should be created by implementing the software architecture using commercially available cloud computing services. We show how to synthesise several existing (healthcare-oriented) paradigms, such as cloud-computing, interoperability, standardisation, and collection and analysis of context-related health data into a requirement-driven reference architecture to support the diagnostic and therapeutic process.

## 2. Background

The German Society for Sleep Research and Sleep Medicine (DGSM) offers comprehensive recommendations regarding sleep diagnostics and therapy. The S3-guideline Insomnia in adultswas most recently updated in 2016 based on new evidence from sleep research [16].

Insomnia is diagnosed by determining up to eight diagnostic criteria defined by the Diagnostic and Statistical Manual of Mental Disorders in its fifth edition (DSM-5): first, anamnesis and status praesens of the patient are obtained. Non-organic insomnia can be already precluded if co-morbid diseases, organic conditions, or (excessive) use of substances interfering with sleep are identified. Next, psychological or psychiatric anamnesis is carried out, excluding non-organic insomnia if the patient suffers from psychological diseases. Then, the patient's individual factors causing insomnia are analysed during sleep anamnesis: using sleep diaries and surveys, sleep behaviour, subjective sleep quality, and

amount of sleep, and occurrences and severity of the sleep disorder will be measured. Sleep diaries are also used during therapy to measure progress and evaluate therapy effects [17].

PSG is currently considered the norm in sleep therapy: patients are continuously and comprehensively monitored for one or more nights. The resulting data, consisting of various biosignals, is evaluated according to a standardised manual [18]. Although PSG can provide exact results, its provision is expensive due to its demand for highly qualified personnel and medical devices.

A more comfortable and less resource-intensive alternative to PSG is actigraphy: recording nocturnal body movements via sensors worn on the body. These movement data can be used to deduce distinct sleep phases, although this intervention usually underestimates the severity of a sleep disorder and overestimates total sleep time [19]. Alternative interventions to accurately and efficiently derive sleep behaviour are continually developed and evaluated: commercially available activity trackers, usually worn around the wrist, do not yet provide sufficiently accurate data, as comparisons with PSG and actigraphy show [5,7]. However, multi-sensor sleep trackers provide results that could still be used to complement subjectively acquired data (e.g., sleep diaries) with objective data. For example, a ring-shaped and hence minimally invasive tracker, provides exact data on sleep onset latency (SOL), total sleep time (TST), and wake after sleep onset (WASO) as data acquired during PSG [20]. As the accuracy of PSG is primarily dependent on ECG data, the development of consumer-grade ECG sensors to be used for in-home sleep analysis promises even more valid results [21,22]. Although studies have shown that environmental and social determinants are related to insomnia as context parameters, they are not officially part of PSG [9–11].

Following an accurate diagnosis, subsequent therapy is the second key factor in successful insomnia treatment. CBT-I is considered the standard therapy method. Pharmacological interventions should only be used if CBT-I is not sufficiently effective or not applicable. Professional recommendations for psychotherapists on the implementation of CBT-I are available [23,24], as well as guides for self-treatment [25,26]. CBT-I attempts to restore the patient's regular sleep routine and includes teaching relaxation techniques and sleep-related information, e.g., correct sleep hygiene. Furthermore, the patient should learn and implement sleep-promoting measures (stimulus control, sleep restriction), as well as cognitive techniques. CBT-I is usually carried out within psychotherapeutic individual or group sessions, resulting in high costs [13]. Speaking of digital cognitive behavioural therapy for insomnia (dCBT-I), most of the aforementioned content and information could be provisioned digitally by means of video-tutorials, questionnaires, step-by-step instructions, and audio recordings. Patients can, thus, control the progress of therapy themselves. The effort for psychotherapists is reduced to training, planning and support activities and the ongoing supervision of patients. dCBT-I has been proven as effective and more efficient than face-to-face therapy forms [27].

The described use case of insomnia therapy is excellent for a a detailed requirement analysis, as well as a reference architecture and implementation making use of cloud technology. The use case underlines the need for: cloud-based data processing; for the use of standards and codes for data exchange and cross-sectional care models; for advanced data analytics based on clinical behaviour; as well as environment data and, of course, health services withdata security and data protection mechanisms. Requirements for future disease-specific cloud-based therapy and diagnosis solutions can be derived from that use case. Health applications depending on health data analysis will benefit from a comprehensive cloud-based infrastructure. Oh et al., for example, provide details on EEG analysis and feature extraction for autism detection [28]. The authors point out a first concept in which EEG data from the patient could be send to a cloud using a smartphone. The paper does not provide any further insight or requirements on the concept, as well as missed important features, such as data standards, clinical codes, and the integration of non-clinical data. Nevertheless, the paper can serve as an input for the concept in this paper. Tang et al. summarise the advantages of AI and ML methods in emergency medicine

and serve as an example of data-based AI analytics in the healthcare context [29]. They point out the advantages which cloud-based services would provide, such as a unified training-inference-visualisation environment which overcomes problems related to the management of the environment, libraries, command-line interfaces and requirements on high-power computing hardware to run AI models. Additionally, they provide an initial concept with different data inputs and even introduce the data input of IoT and wearable sensors. The concept lacks any concrete implementation guideline or reference architecture but is an excellent example underlining the objective of this paper. In the healthcare domain, cloud-based image analysis or storing including a AI based classification is probably one of the most progressive areas. For example, Zhao et al. introduce methods for AI techniques on MRI pictures for Alzheimer's classification [30]. In the discussion by Zhao et al., they mention the advantages of cloud-based processing, mainly referring to the advantage of a larger joint data pool and a related more reliable diagnosis. Interestingly, the authors refer to a publication and suggest that the integration of behavioural data to support the decision making [31]. The very general concept of the cloud setup, not providing any further insight into the technical integration of non-MRI data or the introduction of standards, is presented.

The mentioned papers suggest architectures and concepts to a certain extent. In existing concepts, the purpose, focus and maturity level vary. The systems are used in rehabilitation [32,33], in disease management programs for chronic diseases [34–36], in physiotherapy [37], as well as in behavioural research [38–40]. The purpose of these systems range from support in the implementation of therapy [41] to the monitoring of medical studies [42,43] and the use as a communication medium between patient and HSP [44]. None of the identified systems explicitly proposes a cloud-based reference architecture or provides a systematic development originating from diverse stakeholder requirements. However, there are overlaps in individual aspects, such as the medical field (neurology), area of application (diagnosis and therapy of sleep disorders), and technical implementation (mobile application and cloud platform).

## 3. Methods

The approach chosen for this paper is based on the specifications of the European Standard 62304:2006, chapters 5.2 to 5.4 [45]. This standard defines the software life cycle process for medical devices. The definition of various stakeholder requirements serves as a basis for this work [46]. Stakeholder requirements include aetiology, as well as current diagnosis and therapy methods of non-organic insomnia, i.e., medical-professional requirements. Furthermore, aspects and methods of medical data collection are summarised and possible data sources for medical applications are identified. Moreover, several aspects of the smart city approaches are taken up. Technical requirements are defined based on the selection of data and transmission formats. All stakeholder requirements are identified by integrative literature research. Stakeholder requirements do not exclusively refer to the requirements of the users. The definition of stakeholder requirements is referred to in the standards ISO 13485 (Medical devices—Quality management systems—Requirements for regulatory purposes) and ISO 15288 (Systems and software engineering—System life cycle processes) [46,47].

Based on these stakeholder requirements, specific requirements for the system are derived. In this step, the system is considered as a *blackbox*: initially, no internal components and interfaces are defined. System requirements are divided into categories, including the definition of user interfaces, the behaviour of the system triggered by user input, data formats to be processed and requirements for the installation environment, among others.

Next, the software architecture is drafted. For this purpose, the overall system is divided into software components and atomic software units. Internal interfaces and transmission paths are defined. An overview of the architecture designed in this way is provided as a UML component diagram. Finally, the implementation of the software architecture can be conceptualised. For this purpose, required services for the software com-

ponents are selected and represented as a UML deployment diagram, utilising commercial cloud products.

*Requirements Analysis*

The definitions of stakeholder requirements, as well as the resulting derivation of the system requirements are based upon integrative literature research. This research method does not provide a complete overview of the current state of research. Rather, the analysis and combination of diverse aspects should contribute to the creation of a novel concept [48]. First, the topics and associated keywords, cf. Table 1, were defined. Research was then conducted using the Google Scholar database, using the predefined set of keywords per category. The results were restricted to publications published after 2010. Then, the search results obtained were further restricted to ten publications per category. A publication was only included in the literature collection if the abstract identified thematically appropriate research questions and provided satisfactory answers. In total, 18 sources did not meet this criterion and were, therefore, excluded. This limited selection of publications was then analysed. Secondary references to 66 other publications were included in the bibliography. A total of 18 references to online sources were added manually. All stakeholder requirements are thus to be derived from the pool of 98 full texts and 18 online sources identified. For a schematic representation of the literature search, cf. Figure 1.

**Table 1.** Categories of identified stakeholder requirements and associated search terms.

| Category | Search Terms |
|---|---|
| Diagnosis and therapy | Insomnia, Sleep Disorders, Guidelines, Therapy, Polysomnography |
| Data sources and acquisition | Clinical Outcomes, Internet of Things, Sensors, Wearables, Value-based Healthcare |
| Data formats and standardisation | FHIR, Medical Data Exchange, HL7, Standards, REST |
| Legal aspects | MDR, Medical Device, Standards, GDPR |
| State-of-the-art | Therapy Platform, Clinical Application, Disease Management Programme, mHealth |

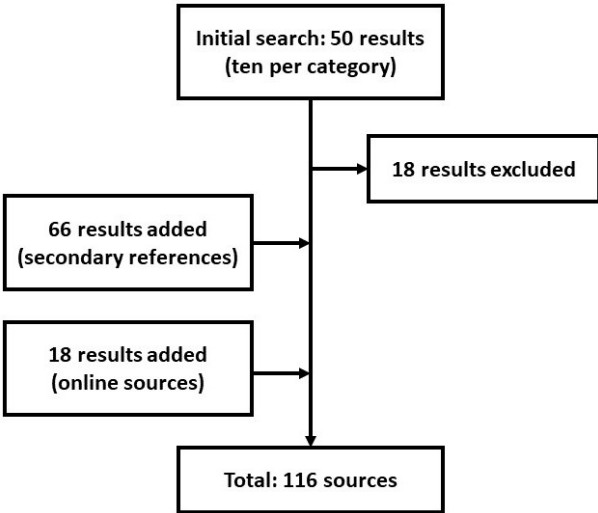

**Figure 1.** Literature research process.

## 4. Requirements

Insomnia depicts the main subject of this paper due to its high prevalence and, compared to other sleep disorders, clearly defined diagnostic criteria. A meta-analysis of over 50 studies shows that insomnia is the most common sleep disorder worldwide. Depending on the definition used, the prevalence ranges from an average of 6–15% of the world population based on an existing diagnosis to an average of 30–48% when defined solely based on subjective criteria, e.g., complications falling asleep or sleeping through the night [49].

Successful treatment would save healthcare costs in the triple-digit million-euro range within six months if the US-based findings of Ozminkowski et al. were applied to the Austrian population [50]. (Average saving per patient over six months: USD 1198 = EUR 985.36 (conversion rate 17.05.2021) × 265,131 (prevalence) = EUR 261,249,482.16) According to Daley et al. most of the financial damage is due to increased sick leave rates and reduced productivity of insomnia patients [51]. The treatment, while being effective and efficient, is hampered by the fact that only a quarter of insomnia patients are correctly diagnosed and treated [52].

### 4.1. Medical Data Collection

Medical diagnosis is based on the measurement of various data that indicate the health status of a person. These measurements also enable the choice and planning of the suitable therapy, adapted to the specific needs of the patient. The measurements and their results are thus regarded as the essence of medicine [53]. Clinical measurement data can be classified according to their sources. A distinction is made at the highest level between patient-reported outcomes (PROs) and non-PROs. PROs are all information that can only be reported by the patient him or herself, without further interpretation by external persons or systems. These data include symptoms, as well as their frequency and severity, and the impact of the disease on the daily life of the patient. PROs are usually acquired using oral or paper-based instruments (e.g., questionnaires), but can also be acquired on a smartphone, tablet or computer, ideally using the ESM/EMA approach (see below). Each PRO-instrument may be checked for validity, reliability, and ability to detect change before clinical application [54].

Observable information, on the other hand, is referred to as non-PROs. Clinician-reported outcomes (ClinROs) describe assessments by medically trained professionals (e.g., doctors, therapists, nurses), while the assessments of laypeople (e.g., caring relatives) are called observer-reported outcomes (ObsROs). Another source is identified by Mayo et al. [55]: technology-reported outcomes (TechROs), originally referred to medical technology-based procedures (functional tests, imaging). However, other data sources, such as mobile applications or IoT devices, are also becoming increasingly important due to their successful use as TechROs, for example, monitoring the intake of medication, measuring physical activity or recording exercise behaviour in seniors [56–58].

### 4.2. Patient-Reported Outcomes: ESM/EMA

Mobile applications are suitable for recording PROs. Defined by the concept of *ecological momentary assessment (EMA)*, PROs should be recorded immediately in the patient's familiar environment [59]. Data, which are collected in this manner, are more reliable than data collected in a retrospective manner, as cognitive biases can be prevented [60]. For example, due to recall bias, respondents do not completely remember past events or sensitivities [61]. Experience sampling method (ESM) is considered as a synonym for the concept of EMA [62]. However, while EMA primarily describes the methodology of data collection, ESM focuses on the aspect of the patient's repeated data collection activities. As both aspects are equally important for data acquisition, the two concepts are often referred to as the combined term ESM/EMA. There are current efforts to explore sleep quality, among other factors, based on the ESM/EMA methodology: Triantafillou et al. use ESM/EMA to investigate the relationship between mood and sleep quality [63], while Li et al. are using the concepts to explore variations in stress levels, sleep quality, and mood in patients suffering from bipolar disorder [64].

### 4.3. Technology-Reported Outcomes

TechROs offer equally important sources of data in that they are the only source that provide objective data compared to all other methods. TechROs include sensors in wearables, mobile devices, applications, and various devices that communicate measured values via the internet under the collective term Internet of Things (IoT). IoT-devices

are already being used successfully in healthcare despite existing technical hurdles [3]. Specifically for the diagnosis of sleep disorders, Fallmann et al. offer an up-to-date, comprehensive overview and evaluation of various methods, indicators, and data sources for sleep analysis [7]. A primary distinction is made between sensors that are worn directly on the body (hence them being called wearables) and devices that are not. Wearables are inexpensive, easy to use, and enable the derivation of various sleep parameters. Data needed for sleep analysis are collected by sensors, typically accelerometers, thermostats or photoplethysmographs.

Devices that are not worn on the body have the advantage of being unintrusive, meaning that patients are not disturbed during sleep. However, the data quality is poorer compared to wearable sensors. Different technologies are used, such as pressure and force sensors, load cells, smartphones, cameras, microphones, and Doppler radar signals. Often several of these sensors are built into one product to increase data quality. For the pure differentiation between waking and sleeping time, motion sensors are best suited. Determining sleep phases, on the other hand, requires additional data that can only be collected by combining several sensors. The measurement of EEG currents using wearables shows this technology's potential [21,65]. However, sensitivity and specificity of PSG are not achieved by any single device or combination of multiple sensors. Fallmann et al. conclude that no difference in data quality will prevail between clinical sleep analysis and home sleep analysis shortly [7]. Similar findings are made by Baron et al., who analyse the use of consumer-oriented wearables and devices for sleep tracking and improvement [6]: the quantified self movement and IoT devices enable the recording and aggregation of granular data. A current issue to be addressed is the lack of standardised, tolerable measurement deviations. Furthermore, proprietary algorithms and the ongoing development of new products impede cross-study comparisons.

### 4.4. Smartphones and Smart Meters

Numerous other data sources lend themselves to supporting the analysis of sleep behaviour. For example, Aledavood et al. are investigating the role of smartphones in sleep tracking of patients suffering from mental illness [66]. Again, it is emphasised that smartphone sensors do not currently match the accuracy of PSG. Nevertheless, the widespread use of smartphones and the foreseeable technical progress emphasise their increasing clinical relevance. A review by Cornet at al. identifies 35 other publications on the usage of smartphones as a data source to enhance health and well-being [67]. The most commonly used sensors include accelerometers (25 studies), location data/GPS (22 studies), call logs (14 studies), and device usage patterns (11 studies).

A novel concept is the extraction of various parameters from a person's electricity consumption. In 2009, the EU Directive on common rules for the internal market in electricity [...] created the basis for the widespread equipping of households with smart electricity meters. As of 2014, 16 member states decided to implement this project after conducting a cost-benefit analysis. The aim was to cover more than 80% of electricity customers by 2020. Wang et al. offer a recent review concerning the analysis of electricity data from smart meters [68]. Wang proves in another publication that socio-demographic data can be derived from the readings of smart electricity meters [69].

In summary, the following requirements regarding medical data collection are identified: ESM/EMA are suitable for optimising the time and content of patients' responses to sleep diaries and sleep questionnaires. This method serves as a tool to collect PROs and will be a central component of the platform. Measured values from IoT devices and wearables enable the collection of objective data. The platform will amass various data sources to derive parameters on sleep quality and duration. These include wearables, smartphone sensor data, IoT devices, and smart electricity meters.

*4.5. Aetiology and Contextual Factors Influencing Insomnia*

In addition to clinical data, several other factors, such as environmental, social, psychological, and chronobiological factors, etc., are associated with different kinds of insomnia. Although these data are usually not tracked in a medical record (EHR, etc.), future applications should take these factors into account, as well as Smart City concepts and factors of urbanisation. By considering these additional data sources, factors that influence diseases can be detected and reduced based on an ICT service infrastructure that facilitates the acquisition of related context information to better diagnose diseases.

According to literature, social factors include higher insomnia rates among divorced, separated or widowed people. In general, prevalence is higher when scholastic level and domestic income are low. Environmental factors include working constraints, noise and air pollution, a damp building environment, etc. [70–72].

Additionally, stress related to socio-affective environmental factors, such as lack of social support or conjugal difficulties, has negative effects on sleep quality. Furthermore, chronobiological factors, such as night working or day–night shifts, promote insomnia by desynchronisation [71].

A study with 1563 participants has shown influencing factors, among medical staff, related to the COVID-19 pandemic, which had underlying social and environmental factors as well. One-third of the medical staff suffered from insomnia symptoms during the COVID-19 outbreak. Factors included the level of education, an isolating environment, and psychological worries about the COVID-19 outbreak and being a doctor [73,74].

The majority of recent findings have suggested that most proposed evolutionary mismatched urban factors are indeed related to the presence of insomnia symptoms. Empirical evidence has indicated that urban inhabitants were more susceptible to developing insomnia symptoms than their rural counterparts. Jiaqing et al. describe evolutionary mismatched factors of urban phenomena which are related to insomnia symptoms and which could be considered in Smart City design: urban environments foster a greater tendency to live alone, reduced support from one's kin due to a more cosmopolitan lifestyle, a less active lifestyle, (more) screen time, less closely knit communities, lesser exposure to natural spaces, greater exposure to artificial light at night, higher level of noise pollution from outdoors and poorer air quality from outdoors; all factors which by evidence promote insomnia-related symptoms [75].

*4.6. Technical Requirements: Data Exchange and Storage*

Regarding the insomnia platform, both medical and administrative data are exchanged and processed. Medical data include personal data of patients, results of EMA/ESM surveys, sensor readings, data from IoT devices and ClinROs of the treating HSP. These heterogeneous data types need to be standardised for improved automated analysis and prediction [76]. Furthermore, a uniform transmission format must be defined for the transmission of the data to the platform. Thus, a uniform endpoint can be created for diverse data types. These two requirements, data structure and transmission format, are met by the FHIR standard of Health Level 7 (HL7) and fulfil the technical requirements of the therapy platform. HL7 is an international Standard Development Organisation and has been involved in the ongoing development of frameworks and associated standards for the exchange of medical data since its foundation in 1987. FHIR, the latest HL7 standard, combines the advantages of existing and widely used standards with those of current web technologies. Although other standards are based on message-based (HL7 V2) or document-based (HL7 CDA) communication, FHIR represents data as independent entities, called resources. A resource could, for example, be a patient, a procedure, a medical problem, or the administration of medication. Resources themselves consist of atomic data fields whose cardinality and structure are defined. Each FHIR resource is uniquely identified and referenced by a UID; referencing between resources is also intentional and prevents duplication of data [77]. Specifying required data fields and value ranges of FHIR resources is called profiling. As an example, Gopinathan et al. use FHIR resources Observation and

Diagnostic Report to represent PROs [78]. In addition, further profiles must be defined for the implementation of the platform based on the following resources, among others:

- Patient for the mapping of the patient;
- Practitioner for the mapping of the HSP;
- Questionnaire for the mapping of the questionnaires and sleep diaries;
- QuestionnaireResponse for the mapping of responses;
- CarePlan for the mapping of the dCBT-I;
- Procedure for the mapping of a therapy step.

For data exchange, the FHIR standard uses the concept of RESTful Webservices, defined interfaces of a webserver via which clients, i.e., requesting systems, can exchange data and call functions of the webserver. FHIR resources are transmitted to a web server via these RESTful application programming interfaces (APIs) and stored there. Resources are queried, updated, and deleted via REST APIs as well [79].

By using FHIR resources, syntactic interoperability between communicating systems can be guaranteed. To ensure semantic, i.e., content-related interoperability as well, the FHIR standard also supports the use of SNOMED-CT, the world's most comprehensive health terminology [77]. Interoperability at process level is driven by the international organisation Integrating the Healthcare Enterprise (IHE): users and manufacturers work closely together to design so-called IHE profiles in which use case-specific processes are mapped in a standardised way. For example, the transfer of TechROs as a FHIR resource is regulated in the recent profile Personal Health Device Observation Upload (POU) and has been officially in test operation since 13 April 2020 [80].

Administrative data include user interface components, treatment processes and other predominantly static data that are rarely, if ever, changed. These data are read by machines but not interpreted, so standardisation is not mandatory. However, to ensure readability and maintainability, Javascript object notation (JSON) is defined as the uniform storage and exchange format for administrative data. JSON is intended to be human and machine-readable within the framework of RESTful web services. Since FHIR can also be mapped to JSON and uses RESTful web services, JSON is preferred in contrast to other formats such as XML or YAML [81].

### 4.7. Smart City Requirements

The following system requirements and possible architecture requirements for health service provision in a Smart City context have been found.

At first the requirement secure service and data access could be identified. Although this requirement seems general for cloud based services, special user access control and role mechanisms might be needed when different services with different data sources are used. De Fuentes et al. suggest the ABC method (attribute-based credentials) to guarantee that a given user is authorised to access a given service or application without leaking any data that are not essential for granting him or her access. In ABC systems, users obtain credentials (specific pieces of information) from an issuer. Each credential contains a set of attributes linked to the user. Based on those credentials, users create presentation tokens that are used to prove the possession of such credentials without disclosing any further information. Privacy features supported by ABC methods are minimal information disclosure, unlinkability of data, key binding, advanced issuance, pseudonyms, inspection, and revocation [82].

For health data storage, as well as health data exchange, most papers suggest the use of EHR or PHRs, as well as clinical decision support system (CDSS) as basic components in a Smart City health service provision [82–84]. As modern health data standards, such as FHIR, can serve as a document as well as message standard, the usage of EHRs or PHRs can be added as a requirement for cloud based therapy and diagnosis platforms.

Clim et al. go even further and suggest a system architecture for clinical decision support systems based on software as medical device (CDS-SaMD) running on smartphones. They define six phases of data processing which are the basic functional requirements

for their proposed architecture. The first phase comprises data acquisition from sensors used by patients. A reference architecture for diagnosis and therapy platform should, therefore, support the integration of IoT and wearable sensor data. The second phase concerns prepossessing of data and transformation to a suitable format that could be usable in the medical field. This requirement links back to the requirement mentioned earlier—to user standards and code systems common in the medical field. As a third step, data are transferred to the central servers operated by the cloud service systems. In a fourth step, the data acquired from cloud services during clinical evaluation is reduced through filtering and extraction methods, which demand the capabilities for flexible cloud-based data analytics and data processing. In the fifth step, visualisation is performed with the help of feature extraction. A requirement is the integration of dashboard tools and possibilities for data visualisations for the professional as well as the patient. As a final and sixth step, the common patterns of diseases and ailments are used for pattern recognition and aid in the overall decision-making process (decisions related to diagnosis and treatment using automatic decision trees and random forest models) [84].

Ouhbi et al. discuss several sustainability requirements, in which sustainability is defined as the extent to which an application is adopted and maintained to achieve durability and continuity in use in a given context. These requirements can be seen as more functional requirements then technical requirements. Economic sustainability aims to maintain capital assets and added value (interest) assets. Individual sustainability refers to the maintenance of the individual human capital, e.g., health, education, skills, and access to services. Individual sustainability can be covered by privacy, safety, security, human–computer interaction, usability, personal health, and well-being. Social sustainability aims to preserve the social capital and preserve services and solidarity of social communities. An example for the environmental dimension of connected health applications is that the app shall reduce transportation means, shall be convenient for frequent use and shall connect to other IT resources. Technical sustainability refers to software systems' longevity and their adequate evolution with changing surrounding conditions and respective requirements [83].

## 5. Results

### 5.1. System Requirements

The system requirements are derived from the previously identified stakeholder and functional requirements. The European Standard 62304:2006 lists twelve categories of necessary software requirement definitions [45]. Software and system requirement can be understood synonymously in the context of this work, as the therapy platform is a software system itself. The standard points out that not all requirements may be known before the product is implemented. Therefore, 41 requirements concerning the five below-mentioned categories were identified in the context of this work. These system requirements enable the subsequent draft of reference architecture and the implementation concept. The requirements are listed in Appendix A.

- Functionality and performance;
- Interfaces between the software system and other systems;
- Data security;
- User interfaces;
- Database definition and requirements.

### 5.2. Software Architecture

Following the European Standard 62304:2006, chapters 5.3 and 5.4, the documentation of the software architecture represents the next logical step in the software development process. The standard requires a subdivision of the platform into software components and units. These units denote the smallest possible subdivision of the software's functionalities. The software units, as well as their interfaces, should be documented in such a way that implementation is possible based on the description. This definition of the internal interfaces thus differs from the definition of the system requirements: defining system

requirements, the system is viewed as a "black box" from the outside, internal details are not apparent. Based on the previously defined system requirements, three software components and 19 software units are identified. An overview of this software architecture can be seen in Figure 2.

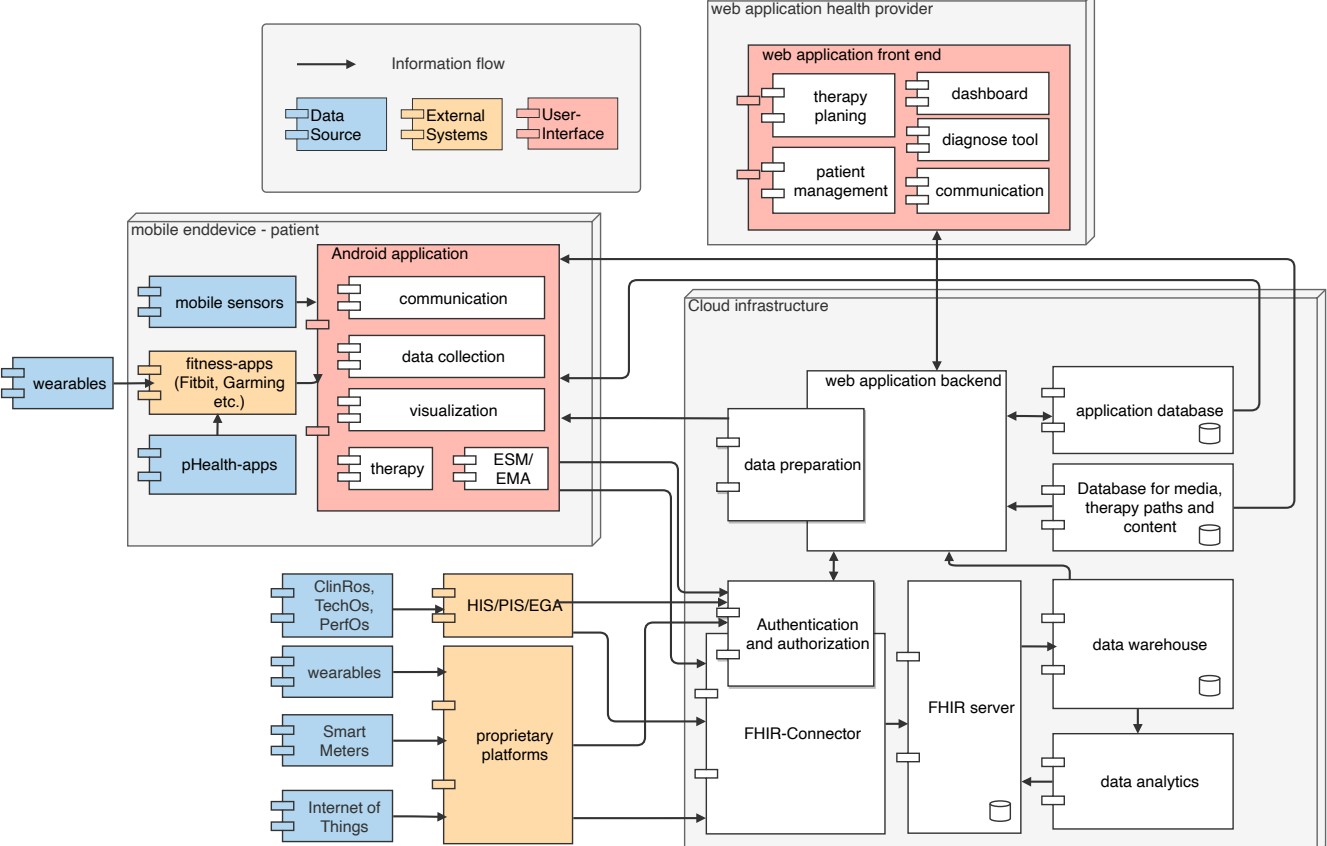

**Figure 2.** UML component diagram of the system architecture.

The component mobile device acts as the user interface for the patient. An installed application gathers various data types using internal, as well as external sensors and ESM/EMA surveys and sends these data to the cloud component. Furthermore, patients communicate with their HSP directly within the app and get access to their personalised therapy plans and information on their sleep quality and behaviour. A HSP accesses the platform via a web application. The application provides an overview on current patients, assists the HSP within the diagnostic and therapeutic process and messaging capabilities to keep in contact with patients.

The central component of the platform is the *cloud environment*: The FHIR server acts as the single source of truth for all health-related data. Data are exported from the FHIR server to a data warehouse and then to a data analytics service, that, in turn, provides recommendations and results as FHIR resources to the FHIR server. The FHIR-connector ensures semantic and syntactic interoperability between external data sources (mobile app, hospital information systems, fitness platforms, IoT-devices, smart meters) and converts and verifies data ingress. An authentication and authorisation component ensures that only entitled users and systems are granted access to the platform. Additional databases are needed to store static and application data. A data preparation component, integrated into the web application backend, extracts and summarises relevant data from the FHIR server, according to either the patient's or HSP's needs.

### 5.3. Implementation Concept

Finally, the software system is implemented after completion and based on the software architecture [45]. The proposed implementation concept comprises the software component cloud environment. Google Cloud Platform (GCP) was chosen as the commercial cloud computing service provider and installation environment. We identified and combined several services from GCP's product range. The therapy platform described in this paper is proposed to be implemented using eleven GCP services, as well as a web and mobile application. An UML deployment diagram of the implementation concept is provided in Figure 3.

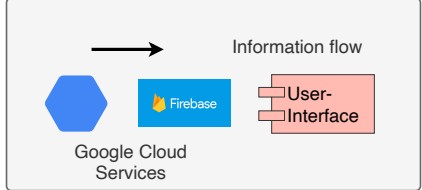

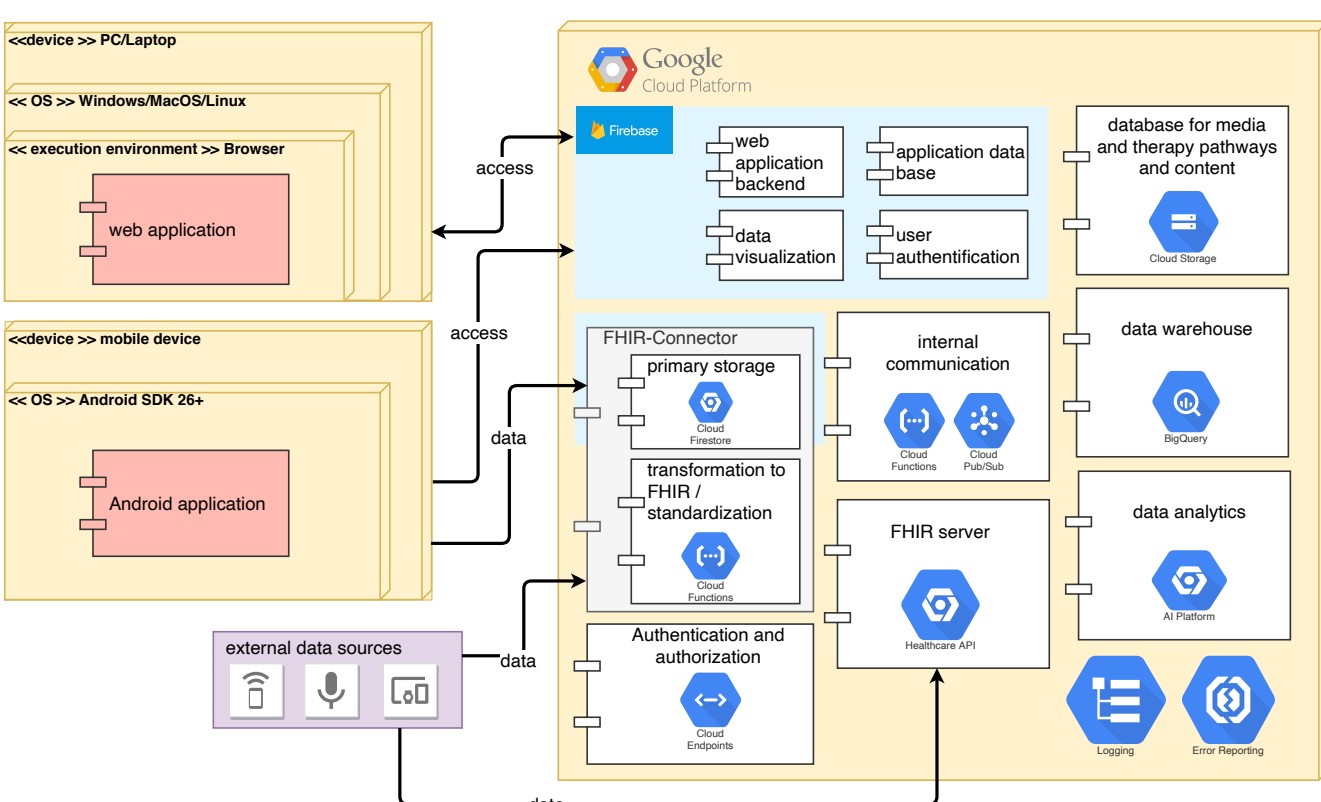

**Figure 3.** UML deployment diagram of the implementation concept.

Service types for storing and analysing data, hosting a web application, as well as authentication and authorisation are needed for the implementation concept. Six system components can be implemented using the multifunctional app-development platform Firebase, providing primary data storage and transformation, user authentication, application backend and application database functionalities. The FHIR server system component can be easily implemented by the GCP service Healthcare API, providing direct access and validation of FHIR resources. Database services used include Cloud Storage (static data), Big query (data warehouse) and Cloud Firestore (primary storage). Business logic for moving and converting data between the services, triggering analysis pipeline and similar jobs are implemented using Cloud Functions, GCP's serverless compute solution.

AI platform, a fully managed, end-to-end platform for data science and machine learning, can be used to yield insights from data stored in the data warehouse.

## 6. Discussion

This paper comprises the structured development process of a digital medical diagnosis and therapy platform based on scientific methods which can be used in future Smart Cities.

The results of an integrative literature research facilitate the identification of stakeholder requirements. These include medical and therapeutic foundations about sleep, sleep disorders, as well as their diagnosis and therapy. Furthermore, organisational, systematic, and technical framework conditions are considered.

These stakeholder requirements are then translated into 41 system requirements. The system requirements, in turn, enable the definition of three software components comprising twelve software units and their internal connections, referred to as software architecture. The last one mentioned, based on the software architecture, proposes an implementation of the system using commercial cloud services.

This paper shows how treatment and diagnosis of a common medical problem (non-organic insomnia) can be supported effectively and economically using modern ICT and cloud computing technology including a context-aware health service provision. The approach chosen is standardised according to the European Standard 62304:2006 and the results of each stage can be traced back to previous findings.

The paper describes a reference architecture for cloud-based health service provision including standardised data storage (FHIR), standardised disease-specific ontology standards (ICD-10), as well as the integration of app-based data collection and a web interface for health provider services. An example implementation based on Google Services is provided as well.

New infrastructures like the implementation of a cloud based health service provision is related to some investment cost. Nevertheless, several advantages when compared to traditional health service provision, can be recognised [85–87]. Consequently, several considerations are discussed.

Akter et al. see the main driver for digitalisation in healthcare in cost reduction. Costs and cloud applications play a vital role in internal management and communication. Cloud solutions can transform healthcare in various ways, such as avoiding unnecessary hospital stays, improving care delivery models using big data analytics, and reductions in costs. As an example, digital transformation in healthcare is said to reduce costs and improve patient outcomes and efficiency, thereby providing a benefit of USD 1.76 billion in Australia [88].

In general, cloud computing helps customers to reduce the hardware, software, and services cost and helps to eliminate software installation and maintenance costs. In an example for implementing electronic health records (EHRs), several other advantages, such as the speed of access to data, improvement of health care, as well as data security can be mentioned [85–87].

Protecting the privacy of data in a cloud environment requires strong security laws. Several mechanism can be applied in the presented reference architecture (such as encryption and decryption concepts such as character-based encryption, unique encryption, public and private key encryption, combination of private and public key encryption, digital signature) and have partly (such as authentication and authorisation) been applied in the presented implementation concept.

A desirable feature of a cloud-based service provision system is to reduce workloads. Scalability can refer to the ability of a system to increase overall performance when adding resources (such as hardware).

The aim is to make services easily available through standard models and protocols without worry about infrastructure, development models, or implementation details.

A key part of the described concept is the use of several data sources coming from wearable devices, mobile endpoints, smart meters, IoT, etc. It is clear that this already

provides new possibilities for clinical decision-making but naturally requires a high degree of data interoperability. Most medical information systems store clinical information about patients in proprietary formats. The suggested solutions make data handling effectively and efficiently by facilitating the retrieval and processing of clinical information about the patient from different locations.

Data analytics capabilities are supported by cloud infrastructure services, as well as standardised collection of data from different sources. As an example, the artificial intelligence medical epidemiology (AIME) (https://aime.life/ last access: 11 October 2021) start-up uses AI-algorithms and data, such as insect-borne diseases, population density, wind speed and direction, rain volume, and other parameters to calculate the outbreak of a disease in a given area. This helped the start-up to predict outbreaks, such as the dengue virus three months in advance. According to Turgeman et al. machine learning on diverse data can lead to estimations on how many days a patient will stay in a hospital which provides the hospital with efficient use of human resources and facilities. Machine learning can, therefore, significantly increase hospital bedding efficiency, thereby enabling a hospital to serve more patients, improve internal staff management and scheduling of resources, and, therefore, support the long-term hospital planning [89].

Overall, the methodology used in this thesis is based on the IEC62304, ISO13485, and ISO15288 standards, but the scope of this work only corresponds to subsections of these standards. The specification of the system architecture is carried out only for the components directly involved in the therapy and diagnosis process. For example, detailed aspects of scaled cloud computing systems, such as elasticity, were not considered in order to provide an easily comprehensible reference architecture. Arani et al. define "elasticity as a crucial feature that distinguishes cloud computing from other distributed computing models. It considers that resource provisioning and allocation processes can be implemented automatically and dynamically" [90]. Therefore, a complete implementation and specification of the architecture is not feasible. The selection of GCP for planning the implementation was not preceded by a market and provider analysis, meaning that other cloud providers are offering comprehensive and possibly more suitable cloud solutions. However, Gartner and Forrester regularly publish analyses of various cloud providers and rate GCP consistently well [91–93].

Despite all limitations, the overall result of this work stands out in terms of approach and results compared to identified existing publications: the entire development process from analysis of requirements to conceptual design and implementation planning is depicted while other publications are limited to only parts of this process.

The findings obtained here could serve as a starting point for the development of a cloud-based market-ready diagnosis and therapy platform in a Smart City context. The standards-oriented approach simplifies a foreseeable certification as a medical device. Furthermore, the applied methodology can be used for the development and evaluation of other diagnostic and therapy systems as well. The following tasks, which are not or only incompletely dealt with in this work, offer the following future research possibilities:

- Determination of user requirements according to the specifications of ISO 9241-110:2020 (Ergonomics of human-system interaction—Interaction principles);
- Clarification of the aspects of data protection and security, as well as ensuring GDPR conformity, cf. [94];
- Complete profiling of FHIR resources;
- Analysis, evaluation, and selection of cloud service providers, cf. [95];
- Implementation of a prototype for the platform, including cloud, mobile, and web component;
- Detailed cost planning;
- Pilot operation and usability testing.

The digitisation of medicine is slowly but steadily progressing. Hence, innovative approaches to solutions are continuously needed. The concept developed in this work, as well as a possible market-ready product based on it, contribute to the modernisation

of healthcare: Cost-cuttings, treatment safety, patient empowerment and value-based healthcare can be achieved and promoted on the basis of current technologies and, thus, change healthcare systems sustainably.

**Author Contributions:** Conceptualisation, D.R.; methodology, D.R. and S.H.; software, D.R.; validation, D.R. and S.H.; formal analysis, D.R.; investigation, D.R.; resources, D.R.; data curation, D.R.; writing—original draft preparation, D.R. and S.H.; writing—review and editing, D.R. and S.H.; visualisation, D.R. and S.H.; supervision, S.H.; project administration, D.R. and S.H. All authors have read and agreed to the published version of the manuscript.

**Funding:** This research received no external funding.

**Institutional Review Board Statement:** Not applicable.

**Informed Consent Statement:** Not applicable.

**Data Availability Statement:** Not applicable.

**Acknowledgments:** This research was partially supported by Solgenium OG, offering us resources to facilitate literature research. We especially thank our colleague Matthias Gira who provided insight and expertise that greatly assisted the research process.

**Conflicts of Interest:** The authors declare no conflict of interest.

## Abbreviations

The following abbreviations are used in this manuscript:

| | |
|---|---|
| API | Application Programming Interface |
| AWMF | Arbeitsgemeinschaft der Wissenschaftlichen Medizinischen Fachgesellschaften |
| (d)CBT-I | (digital) Cognitive Behavioural Therapy for Insomnia |
| CDSS | Clinical Decision Support System |
| DGSM | Deutsche Gesellschaft für Schlafforschung und Schlafmedizin |
| DSM | Diagnostic and Statistical Manual of Mental Disorders |
| EHR | Electronic Health Record |
| EMA | Ecological Momentary Assessment |
| ESM | Experience Sampling Method |
| FHIR | Fast Healthcare Interoperability Resources |
| GCP | Google Cloud Platform |
| GDPR | General Data Protection Regulation |
| HSP | Health Service Provider |
| ICD | International Statistical Classification of Diseases and Related Health Problems |
| IHE | Integrating the Healthcare Enterprise |
| MDR | Medical Device Regulation |
| ÖGSM | Österreichische Gesellschaft für Schlafmedizin und Schlafforschung |
| PHR | Personal Health Record |
| PROs | Patient Reported Outcomes |
| REST | Respresentational State Transfer |
| SaMD | Software as Medical Device |
| SOL | Sleep Onset Latency |
| TechROs | Technology Reported Outcomes |
| TST | Total Sleep Time |
| WASO | Wake After Sleep Onset |

**Appendix A. System Requirements**

*Appendix A.1. Requirements for Functionality and Performance*

1.  The system is operated entirely in the cloud by using commercially available services of a cloud provider. Exception: The access point for patients is a native mobile application.

*Appendix A.2. Interfaces between the Software System and Other Systems*

2.  The system provides a REST interface to submit FHIR resources to the system after successful authorisation;
3.  The system provides a generic API for the transmission of proprietary data formats for other data sources that do not have FHIR functionality;
4.  The system can send requests to Google's Fused Location Provider API to retrieve the user's location data;
5.  The system can make requests to Google's Fit API and thus access activity and lifestyle data;
6.  The system can call the Android system methods BatteryManager and *UsageStatsManager* to access battery and usage data;
7.  The system receives further data via a public REST API.

*Appendix A.3. Requirements for Data Security*

8.  The system allows only authorised patients to access the Android application. Users can register themselves and are confirmed by the HSP. Registration is completed by entering an e-mail address and password. Only then can the data collection be started;
9.  The system only allows authorised HSPs to access the web application;
10. User accounts for HSPs are created and activated by an administrator;
11. The system only allows authorised systems to transfer data to the system. Authorisation is completed using OAuth 2.0;
12. The system has functions that allow all administrative activities and accesses to be logged.

*Appendix A.4. User Interface Requirements Implemented by Software*

13. The system provides a method for authentication and authorisation of the patient via email and password;
14. The system allows access to other functions only when the user has been successfully authorised and confirmed;
15. The system allows the patient to activate and deactivate the collection of personal data depending on the type of data;
16. The system only collects and transmits personal data if the user agrees to the collection and the collection of the respective data has been activated;
17. The system allows users to communicate with the relevant HSP via a secure communication channel;
18. The system offers functions that enable the patient to carry out a digital CBT-I on his or her own responsibility. For this purpose, the system accesses predefined therapy procedures that are assigned by the HSP;
19. The system logs the user's therapy progress;
20. The web application allows a HSP to authenticate and thus log in to the system via email and password;
21. The system clearly displays all users of a HSP in a list;
22. The system provides an overview of a patient's data. This includes status (registered, confirmed, data collection started—diagnosed—in therapy—therapy completed—treatment episode completed), personal data and key figures based on collected data (ClinROs, PROs, TEchROs);
23. The system allows the HSP to view all raw data collected;
24. The system enables the HSP to exchange messages with the patient via a secure communication channel;

25. The system provides an overview of the diagnostic criteria for non-organic insomnia according to DSM-5 and implements a workflow based on the five diagnostic steps. The system is used exclusively for documentation;
26. The system makes it possible to assign a patient to a previously defined therapy sequence;
27. The system makes it possible to ask patients to fill in questionnaires or sleep diaries, either once or at regular intervals;
28. The system allows to inform an administrator about problems that occur.

Administrative Interface

29. The system can be manually adjusted by administrators;
30. For manual customisation, tools from the selected cloud provider can be used if available. No additional user interface needs to be designed;
31. System administrators can create and manage accounts for HSPs;
32. If available, the identity access management tool of the cloud provider must be used. Roles and permissions are to be assigned according to the principle of minimal rights.

*Appendix A.5. Data Definition and Database Requirements*

33. FHIR profiles are specified against FHIR resources for mapping the data;
34. The system can receive inhomogenous raw data and convert it into standardised FHIR resources (version R4);
35. The system receives and stores FHIR resources (version R4) on a FHIR server;
36. The system stores CBT-I media content in a database;
37. The system stores textual content of the CBT-I and other static data in JSON format;
38. The system stores user settings and other details in a database;
39. The system implements an authorisation system to prevent unauthorised access to external data;
40. The system transfers standardised FHIR resources to a data warehouse;
41. The system sends calculation results of the machine learning component to the FHIR server.

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
