# Peer review of "Requirements and Architecture of a Cloud Based Insomnia Therapy and Diagnosis Platform: A Smart Cities Approach"

_smartcities, doi:10.3390/smartcities4040070_

Round 1

Reviewer 1 Report

  1. The authors should elaborate in the abstract on the comparison of thier work with which one from the literature. 
  2. The authors need to list all contributions clearly in Sec1.
  3. The use of cloud as a buffer needs more coverage in the related work. For example, the authors in https://ieeexplore.ieee.org/abstract/document/8784043 a buffer/storage management that controls the incoming requests to the cloud. Have you considered the request to the cloud in your work? 
  4. Sec4.1 is short and needs further elaboration about the approach to collect the data. 
  5. In Sec4.7, the requirements are not clear. What are they? can you list them clearly?
  6. Sec4.8 is about system requirement, and Sec4.7 about the system requirement in smart cities. These 2 sections can easily be merged for clarity. 
  7. Please add some results and critical discussion to the paper.

Author Response

Cover Letter

Manuscript ID: smartcities-1390404

Title: Requirements and Architecture of a Cloud Based Insomnia Therapy and 

Diagnosis Platform: A Smart Cities approach

Authors: Daniel Reichenpfader *, Sten Hanke

Received: 6 September 2021

Dear Reviewers,

The authors are thankful for the valuable comments provided. We reworked the manuscript and tried to cover all points raised. We certainly hope the revision is now of higher quality and value. 

Please see below where and how the comments have been addressed.

Besides the below mentioned changes, the paper was proofread by a professional agency. 

These language-related changes have not been tracked by the TrackChanges-package due to limitations in Latex.

[line numbers]

Reviewer 1: 

  1. The authors should elaborate in the abstract on the comparison of their work with which one from the literature.

An analysis of existing systems was added to the abstract [19-20] and to section 2 [186-196] .

  1. The authors need to list all contributions clearly in Sec1.

Main contributions added in Sec1 [78-87]

  1. The use of cloud as a buffer needs more coverage in the related work. For example, the authors in https://ieeexplore.ieee.org/abstract/document/8784043 a buffer/storage management that controls the incoming requests to the cloud. Have you considered the request to the cloud in your work? 

Cloud computing elasticity as well as a complete technical specification are not the primary focus of our paper. However, in order to secure future consideration, the topic was added in Sec6. [642 - 646]

  1. Sec4.1 is short and needs further elaboration about the approach to collect the data. 

Added details on acquisition of patient reported outcomes. [273 - 278]

  1. In Sec4.7, the requirements are not clear. What are they? can you list them clearly?

Chapter has been reworked and the smart cities related requirements for cloud based diagnosis and therapy systems better defined.

BIM requirements has been removed because not relevant.

  1. Sec4.8 is about system requirement, and Sec4.7 about the system requirement in smart cities. These 2 sections can easily be merged for clarity. 
  2.  

System requirements have been moved to results to make more clear the process and one achievement of the paper → the development of the reference architecture based on the 41 system requirements derived from several stakeholder and functional requirements.

Requirements have been added to the ANNEX.

  1. Please add some results and critical discussion to the paper.

An enumeration of the mentioned system requirements was added in the appendix and referred to in Se5.1 [512].

Comparison of systems has been added and the discussions extended

Changed “Smart City system requirements” to “Smart city requirements” to distinguish better between system requirements which represent a distinct step in the process.

Reviewer 2: 

  1.   In the abstract, “However, a successful implementation requires the consideration of additional aspects including detailed cost planning, methods for data analysis as well as data security and safety.”, can authors provide this information for their system in this manuscript? If they cannot provide this information, then their implementation is not successful according to their own statement. This is an important consideration.

The paper described a reference architecture for cloud-based health service provision including standardized data storage (FHIR), standardized disease-specific ontology standards (ICD-10) as well as the integration of app-based data collection and a web interface for health provider services. 

An example implementation based on Google Services is provided as well. 

Data security is supported by the suggested implementation and through the use of standard conform cloud infrastructures (see https://cloud.google.com/security). Other cloud providers provide similar data security mechanisms and can be applied following the suggested reference architecture in the paper. Of course individual specification and requirements need to be checked but in general in most cases certifications by the cloud infrastructure provider are standard. In any case the data security in the cloud can be seen as equal or even better compared to data storing in hospital environments.

Data analytics capabilities are supported by the provided reference architecture through the standardization of data, the use of health, behaviour and context data as well as the comprised electronical storage of these data per se. 

Cost factors of the introduction of cloud infrastructure have been provided as well as the benefits and the cost savings on long term presented. The abstract has been adapted as well a new chapter (5.3) introduced summerizing the discussion.

  1.   Is there any data to support the authors’ system architecture? The current analysis is mainly on previous literature.

Yes the concept is a reference architecture combining several aspects which are i) cloud infrastructure based services and health data collection, ii) the introduction of a data interoperability concepts by using FHIR, iii) the introduction of a international coding (ICD-10) which is already used in clinical practice and a requirements (as suggested by the authors) for a service provision along the continuity of care and a intersectoral care and iv) the introduction of context related data for health data analytics. All the concepts might be introduced by its own and have been proven to provide chances and advantages, nethertheless so far no concept has been implemented making use of all the presented aspects. Implementations of the different aspects on his own have been referenced. An implementation of the whole concept comprising all aspects is still pending. The papers purpose has been to provide such a concept based on requirements.

  1.   Have there been other similar systems like this? The authors can compare their system with others to illustrate the strength or shortcomings.

Paragraph has been added to Background/ Sec2 [186-196].

None of the identified systems explicitly proposes a cloud-based reference architecture nor provides a systematic development originating from diverse stakeholder requirements. However, there are overlaps in individual aspects, such as the medical field (neurology), area of application (diagnosis and therapy of sleep disorders) and technical implementation (mobile application and cloud platform). 

  1.   The literature review related to cloud based is too terse. Maybe the authors can take a leaf from some of the medical / healthcare domains. New literatures can be added to improve the literature review, which is beneficial for readers to know this area, such as the cloud-based system 10.1007/s40747-021-00408-8

https://doi.org/10.1016/j.bbe.2021.02.006

https://doi.org/10.1016/j.bbe.2020.12.002

“A novel automated autism spectrum disorder detection system” by Oh et al. is a very good example and use case which could make use of the presented submission. Clearly the paper is excellent and summerizes several approaches to detect autism. In any case the authors themself refer in their discussion that a cloud-based approach could lead to a faster diagnosis and is considered as next step for the authors (providing other advantages as also suggested by the paper under review). Additionally the dimension of using standards (for data exchange) as well as contextual data could be a benefit for the autism diagnostics and service provision. 

“Artificial Intelligence and Machine Learning in Emergency Medicine” by Tang et al. is an additional excellent paper describing the chances to use of AI/ ML methods in Emergency Medicine. The paper is as well in chapter 3.3 underlying the authors concept as Tang et al. are referring to cloud-based solutions in future which will provide better AI outcome and the possibility to extend the cloud environment by tools for better disease diagnosis and predictions. The draft of a concept in figure 3 in Tang et al. can indeed serve as an input to the concept provided by the authors of the paper under review.

“Application of Artificial Intelligence techniques for the detection of Alzheimer’s disease using structural MRI images” by Zhao et al. is as well providing a good example use case for the proposed solution. In the discussion the authors refer to future cloud solutions and as well interestingly referring to the point of using context (or non clinical data) for enhancing diagnosis. This is exactly the point the authors of the paper under review suggest. The paper is an excellent input reference for the proposed reference architecture.

Papers have been used to add details to the background chapters (see lines 147 ff). The authors thank for the interesting insides and paper suggestions.

  1. Overall, i think the objective of the this work should be fleshed out even more clearly in the introduction. Please re-work on your wording and emphasize the objectives. 

Clarified objectives in Sec1. [48-87]

Reviewer 2 Report

1.    In the abstract, “However, a successful implementation requires the consideration of additional aspects including detailed cost planning, methods for data analysis as well as data security and safety.”, can authors provide this information for their system in this manuscript? If they cannot provide this information, then their implementation is not successful according to their own statement. This is an important consideration.

2.    Is there any data to support the authors’ system architecture? The current analysis is mainly on previous literature.

3.    Have there been other similar systems like this? The authors can compare their system with others to illustrate the strength or shortcomings.

4.    The literature review related to cloud based is too terse. Maybe the authors can take a leaf from some of the medical / healthcare domains. New literatures can be added to improve the literature review, which is beneficial for readers to know this area, such as the cloud-based system 10.1007/s40747-021-00408-8

https://doi.org/10.1016/j.bbe.2021.02.006

https://doi.org/10.1016/j.bbe.2020.12.002

5.  Overall, i think the objective of the this work should be fleshed out even more clearly in the introduction. Please re-work on your wording and emphasize the objectives. 

Author Response

(The authors gave the same response as above.)

Round 2

Reviewer 1 Report

The authors addressed all the comments properly and I have no other comments for this paper. I recommend accepting the paper for publication in the present form. 

Reviewer 2 Report

the revised manuscript can be accepted now.